# PD-L1-Targeted Co-Delivery of Two Chemotherapeutics for Efficient Suppression of Skin Cancer Growth

**DOI:** 10.3390/pharmaceutics14071488

**Published:** 2022-07-18

**Authors:** Fatemeh Movahedi, Jie Liu, Bing Sun, Pei Cao, Luyao Sun, Christopher Howard, Wenyi Gu, Zhi Ping Xu

**Affiliations:** Australian Institute for Bioengineering and Nanotechnology, The University of Queensland, St. Lucia, QLD 4072, Australia; f.movahedi@uq.edu.au (F.M.); j.liu5@uq.net.au (J.L.); bing.sun@uq.net.au (B.S.); pei.cao@uq.edu.au (P.C.); luyao.sun@uq.net.au (L.S.); c.howard2@uq.edu.au (C.H.); w.gu@uq.edu.au (W.G.)

**Keywords:** lipid-coated calcium phosphate nanoparticles, antiworm albendazole, TOPK inhibitor OTS964, combination cancer therapy, dual targeting delivery, programmed death ligand-1 targeting

## Abstract

To overcome the severe side effects of cancer chemotherapy, it is vital to develop targeting chemotherapeutic delivery systems with the potent inhibition of tumour growth, angiogenesis, invasion and migration at low drug dosages. For this purpose, we co-loaded a conventional antiworm drug, albendazole (ABZ), and a TOPK inhibitor, OTS964, into lipid-coated calcium phosphate (LCP) nanoparticles for skin cancer treatment. OTS- and ABZ-loaded LCP (OTS-ABZ-LCP) showed a synergistic cytotoxicity against skin cancer cells through their specific cancerous pathways, without obvious toxicity to healthy cell lines. Moreover, dual-targeting the programmed death ligand-1 (PD-L1) and folate receptor overexpressed on the surface of skin cancer cells completely suppressed the skin tumour growth at low doses of ABZ and OTS. In summary, ABZ and OTS co-loaded dual-targeting LCP NPs represent a promising platform with high potentials against complicated cancers where PD-L1/FA dual targeting appears as an effective approach for efficient and selective cancer therapy.

## 1. Introduction

Cancer is one of the leading causes of death [1]. At the current stage, cancer chemotherapy is still one of the main treatment streams, while it is challenged by severe side effects, such as nausea, vomiting, diarrhoea, fatigue and oral mucositis [2]. This challenge necessitates the development of effective chemotherapeutic systems using low dosages of safe drugs. To this end, the first requirement is that selected drugs should be only toxic against cancer cells but have minimal side effects on healthy tissues. Therefore, repurposing safe conventional drugs (such as antiparasitic drug albendazole) with anticancer mechanisms is a promising approach.

To improve the anticancer efficacy of such safe drugs, a suitable delivery system is important. Previously, lipid-coated calcium phosphate (LCP) core-shell nanoparticles (NPs) were used to encapsulate for delivery of drugs [3], genes [4] and vaccines [5]. Biocompatible calcium phosphate (CaP) nanomaterials are approved by the FDA for human use [6], and the pH sensitivity triggers CaP dissolution in the acidic tumour microenvironment and endosome/lysosome for enhanced therapy [7,8]. The enhanced efficacy has also been demonstrated in recent reports. For example, hydrophobic albendazole (ABZ) and alpha-tocopheryl succinate (α-TOS) were loaded as tiny crystals in the core (CaP) and in the lipid bilayer of LCP nanoparticles, respectively, significantly enhancing anti-cancer efficacy in vitro and in vivo [9,10]. As demonstrated as an excellent delivery system, LCP nanoparticle was selected in this research. 

Even though encapsulating the safe drug (ABZ) in LCP nanoparticles significantly increases its cytotoxicity against cancer, e.g., decreasing IC50 from 20 (free ABZ) to 2.5 µg/mL (ABZ-LCP) for B16F0 cells, the inhibition of cancer growth in vivo was only 60% at the ABZ dose of 5 mg/kg [10], due to the mild anti-cancer activity of the safe drug ABZ. To further enhance the cancer therapy, combining another drug that has a mechanism of action complementary to ABZ is necessary. Thus, the current research developed combination strategy for synergistic cancer chemotherapy by incorporating a widely used antiparasitic drug, albendazole (ABZ) and a T-LAK cell-originated protein kinase (TOPK) inhibitor, OTS964, in LCP nanoparticles in the core and lipid bilayer (OTS-ABZ-LCP NPs), respectively (Figure 1). Albendazole (ABZ) inhibits tubulin polymerisation and angiogenesis [11], and OTS964 (OTS) is a recently developed inhibitor of TOPK [12], whose overexpression is associated with poor clinical outcomes [13]. Both ABZ and OTS964 induce generation of reactive oxygen species and damage the mitochondria (Figure 1). 

Recently, LCP nanoparticles are used for combination therapy by co-delivering chemotherapeutics with different mechanisms of action. One major advantage of CaP nanomaterials is pH sensitivity, which triggers CaP dissolution in the acidic tumour microenvironment and endosome/lysosome for enhanced delivery and therapy [7]. Similarly to liposomal systems, the lipid bilayer on the surface of CaP nanoparticles can be used to carry hydrophobic drugs (such as α-TOS [8] and OTS964); however, LCP NPs are preferred to conventional lipid nanosystems since the CaP core provides a matrix for encapsulation of hydrophobic drugs as very tiny nanocrystals of <3–5 nm for improved bioavailability of those drugs [10]. In addition, Jie et al. employed LCP NPs for co-delivery of cell death-siRNA and indocyanine green for combined gene/photothermal therapy [14]. 

In this research, we employed lipid-coated calcium phosphate (LCP) nanoparticles (NPs) for co-delivery of ABZ and OTS964 for synergistic combination chemotherapy. As shown in Figure 1, ABZ nanocrystals are presumably embedded with the CaP matrix [10], and hydrophobic OTS964 is loaded in the lipid bilayer, which synchronises the induction of cancer cell apoptosis. 

To minimise the impact on healthy tissues, we have developed a dual targeting strategy to specifically deliver to the skin tumour tissues based on the fact that skin cancer cells overexpress programmed death ligand-1 (PD-L1) [15] and folate receptors [16] as surface proteins. While targeting PD-L1 overexpressed on many tumour cells [17] is widely reported for cancer immunotherapy with PD-L1 antibody [18], targeting this ligand for chemotherapeutic delivery is reported very rarely. Elegantly, a bispecific antibody, i.e., anti-PD-L1-linker-anti-OCH_3_, was specifically designed, prepared and conjugated onto the OTS-ABZ-LCP NP surface via specific interactions between anti-OCH_3_ antibody and OCH_3_-PEG-lipid (Figure 1) [19]. Furthermore, folic acid (FA) was conjugated via FA-PEG-lipid incorporation into the lipid bilayer to endow LCP NPs with dual targeting capability [20], which can be readily prepared in the laboratory. 

Our in vitro experiment data demonstrate that the combination therapy was much more effective against melanoma cells through complementary mechanisms of action but highly biocompatible with healthy cell lines. Moreover, treatment with dual targeting OTS-ABZ-LCP NPs nearly completely inhibited B16F0 tumour growth at very low drug dosages (2.5 mg/kg ABZ and 1.0 mg/kg OTS964 for 3 times) in the mouse model. The dual targeting OTS-ABZ-LCP NPs show high promise as a biocompatible delivery platform for synergistic combination cancer therapy. 

## 2. Material and Methods

### 2.1. Materials

Albendazole (ABZ), calcium chloride, sodium phosphate, cholesterol and igepal-CO520 were purchased from Sigma-Aldrich (St. Louis, MO, USA). Cyclohexane was purchased from Merck (Darmstadt, Germany). MTT (3-(4,5-Dimethylthiazol-2-yl)-2,5-diphenyltetrazolium bromide) was supplied by Invitrogen (Carlsbad, CA, USA). OTS964 was purchased from Chimietek (Indianapolis, IN, USA), and 1,2-dioleoyl-sn-glycero-3-phosphate (DOPA), 1,2-dioleoyl-sn-glycero-3-phosphocholine (DOPC), 1,2-distearoyl-sn-glycero-3-phosphoethanolamine-N-[folate(polyethylene glycol)-2000] (DSPE-PEG(2000)-FA) and 1,2-dipalmitoyl-sn-glycero-3-phosphoethanolamine-N-[methoxy(polyethylene glycol)-2000] (DSPE-PEG(2000)-OCH_3_: mPEG) from Avanti Polar Lipids. H_2_DCFDA was obtained from Promokine (Heidelberg, Germany), apoptosis analysis kit from Biolegend (San Diego, CA, USA) and VEGF ELISA kit from Abbkine (Shanghai, China). PD-L1 bispecific antibody was expressed, at the University of Queensland, as previously reported [21]. Human umbilical vein endothelial cells (HUVEC) and B16F0 cells were obtained from American Type Culture Collection (ATCC, Manassas, VA, USA). All other reagents and chemicals were purchased from Sigma-Aldrich. Milli-Q water was used in all experiments.

### 2.2. Fabrication and Characterisation of LCP-Based Nanoparticles

Albendazole-loaded LCP (ABZ-LCP) nanoparticles were fabricated as described previously [22]. In brief, an aqueous solution containing CaCl_2_ and ABZ was dispersed in 5 mL of cyclohexane/Igepal CO-520 (7/3, *v*/*v*), forming a well-dispersed microemulsion, and mixed with a similar microemulsion of alkaline Na_2_HPO_4_ with 50 µL of DOPA (20 mM in chloroform). After stirring for 20 min, the resultant CaP-DOPA cores were collected in absolute ethanol and washed thrice. CaP-DOPA cores were redispersed in 1 mL of chloroform. Then, 31.25 µL of DOPC (20 mM) and cholesterol (20 mM) in chloroform was added and mixed, and chloroform was evaporated under the vacuum to form a thin layer film. The film was hydrated with PBS (pH 7.4) and gently sonicated. Similarly, Cy5-dsDNA-loaded LCP NPs (Cy5-LCP) were synthesised by replacing ABZ with Cy5 dsDNA.

To synthesise OTS964-loaded LCPs (OTS-LCP or OTS-ABZ-LCP), OTS964 was dissolved in chloroform, and mixed with DOPC and cholesterol chloroform solution. The chloroform solution was then used to collect the NPs, followed by evaporating the chloroform under the vacuum. The resultant film was then hydrated with PBS. The amount of loaded ABZ and OTS was determined by dissolution of LCPs in lysis buffer, centrifugation at 20,000× *g* for 15 min, and the absorbance of the supernatant at 290 nm (for ABZ) and 225 nm (for OTS), respectively. 

To conjugate folic acid to LCP nanoparticles, 5–20% DOPC of the outer lipid layer was replaced with DSPE-PEG(2000)-FA (20 mM). To attach PD-L1 to LCP nanoparticles, DOPC was replaced with 10% DSPE-PEG(2000)-OCH_3_ (mPEG), and the resultant LCP NPs were mixed with 1.1 mg/mL bispecific PD-L1 antibody (bi-anti-PD-L1) solution via specific interactions with CH_3_O in DSPE-PEG [23]. The nanoparticles were then purified by centrifugation at 20,000× *g* for 30 min along with ultrafiltration using 100 kDa membrane to remove unconjugated ligands. The full list of various synthesised NPs is presented in a Appendix A.

Nanoparticles were characterised by determining the hydrodynamic diameter and the zeta potential in a Zetasizer (DLS, Zetasizer Nano, Malvern, UK). The images were taken in a transmission electron microscope (TEM, JEM-3010, ZEOL, Tokyo, Japan). 

### 2.3. Determination of the Number of Folic Acid and Bi-Anti-PD-L1 Conjugates

The concentration of folic acid was determined by measuring the absorbance at 285 nm. The concentration of conjugated bi-anti-PD-L1 was determined by subtracting the amount of unbonded antibody in the supernatant from the initial antibody amount. The amount of bi-anti-PD-L1 in the supernatant was measured in a Nanodrop spectrophotometer. The number of unconjugated folic acid or bi-anti-PD-L1 per LCP NPs was calculated as described in Appendix A. 

### 2.4. Quantification of Surface PD-L1 Expression

The level of PD-L1 expression on B16F0 cells was determined by flow cytometry. The cells at the density of 5 × 10^4^ cell/well were seeded in 12-well plates for 24 h. Then, the cells were detached by trypsinisation (0.25% trypsin) and centrifuged at 400× *g* for 5 min. The cell suspension was then mixed with PE antimouse PD-L1 antibody (BioLegend, San Diego, CA, USA) and incubated for 30 min at room temperature. The cells were then washed and resuspended in flow cytometry staining buffer (PBS supplemented with FBS) and analysed by flow cytometry (BD Accuri C6 Flow Cytometer Thermo Fisher Scientific, Sydney, Australia).

### 2.5. Cellular Uptake of Bi-Anti-PD-L1-Conjugated LCP NPs

The cellular uptake of bi-anti-PD-L1 conjugated LCP NPs containing Cy5 dsDNA was determined by flow cytometry. B16F0 cells in Dulbecco’s Modified Eagle Medium (DMEM) at the density of 5 × 10^4^ cell/well were seeded in 12-well plates overnight to about 70% confluency. After 24 h, DMEM was replaced with a medium containing Cy5-LCP NPs ([Cy5] = 25 nM) conjugated with 0, 40, 80 or 160 bi-anti-PD-L1 per LCP. After 4 h incubation, media were removed and washed with PBS thrice. Then, the cells were detached by trypsinisation and resuspended in flow cytometry staining buffer to determine the cell fluorescence using flow cytometry (BD Accuri C6 Flow Cytometer).

To block the receptors, partial cells were incubated with media containing 5 µM bi-anti-PD-L1 for 2 h. The media were then replaced with fresh media containing Cy5-LCP-P160 NPs, followed by similar post treatment. 

### 2.6. Cytotoxicity Studies

Cytotoxicity of blank LCP, free ABZ, free OTS, ABZ-LCP, OTS-LCP and OTS-ABZ-LCP formulations against B16F0 and HUVEC cell lines was assessed using 3-(4,5-dimethylthiazol-2-yl)-2,5-diphenyltetrazolium bromide (MTT) assay. Briefly, cells in DMEM supplemented with 10% FBS and 1% penicillin-streptomycin were seeded in a 96-well plate (4000 cell/well) at 37 °C under 5% CO_2_ via overnight incubation. Then, media were replaced with fresh media containing different formulations of ABZ and/or OTS. After 24-h incubation, media were changed with fresh media containing 10% MTT (5 mg/mL), and cells were incubated for another 4 h. After removal of media, each well was added with 50 µL of dimethyl sulfoxide (DMSO). After shaking for 10 min, the absorbance at 570 nm was recorded, and the viable cell percentage calculated based on the absorbance in relation to the control experiment. 

To determine cytotoxicity by folic acid/bi-anti-PD-L1-conjugated ABZ-OTS-LCP NPs, B16F0 cells were seeded in 96-well plates at the density of 4000 cells per well overnight. Then, media were replaced with that containing ABZ-OTS-LCP NPs (equivalent to 100 ng/mL of ABZ and 50 ng/mL of OTS964) conjugated with different concentrations of folic acid (50, 100 or 200 ligands per LCP), bi-anti-PD-L1 (40, 80 or 160 ligands per LCP) or their combinations. Cells were incubated with LCP NPs for 24 h, and then media were removed. Cells were washed with PBS two times and then augmented with media containing 0.5 mg/mL MTT. After incubation for 4 h, media were removed and replaced with 50 µL of DMSO, whose absorbance at 570 nm was determined by spectrophotometer after shaking carefully. 

For the blockage of receptors, cells were incubated with media containing 1 mM of folic acid or 5 µM of bi-anti-PD-L1 for 2 h. Then, media were replaced with that containing OTS-ABZ-LCP-F100 or OTS-ABZ-LCP-P160, followed by similar post treatments. 

To evaluate drug synergism, [A] × [B]/[A + B] > 1.2 was considered as synergistic [24], where [A] represents the viability of cells upon treatment with ABZ-LCP, [B] the cell viability upon treatment with OTS-LCP, and [A + B] the cell viability upon OTS-ABZ-LCP treatment.

### 2.7. Analysis of Apoptosis Induction, Reactive Oxygen Species (ROS) Generation and VEGF Secretion

To quantify the percentage of apoptotic and necrotic cells, treated cells were harvested after 24 h treatment with LCP formulations containing ABZ, OTS, or OTS-ABZ. Then, cells were washed thrice with PBS and collected by 0.25% trypsin, followed by centrifugation at 2000 rpm for 5 min. The resultant cell pellet was stained with Annexin V–FITC and propidium iodide (PI) kit (BioLegend, San Diego, CA, USA) according to the supplier’s protocol. Briefly, the pellet was resuspended in 100 µL of binding buffer, followed by adding 5 µL of FITC Annexin V and 10 µL of propidium iodide (PI) solution. Then, the cells were dark-incubated for 15 min, and 400 µL of Annexin V binding buffer was added to each sample. The fluorescence of these stained cells was measured in a flow cytometer (CytoFLEX, Beckman, CA, USA), and the data processed with CytExpert software (Brea, CA, USA).

ROS level was also determined using H_2_DCFDA. After treating the cells with ABZ-LCP, OTS-LCP or OTS-ABZ-LCP for 24 h, media were collected. The cells were further incubated in fresh media with 5 µM of H_2_DCFDA for 30 min and collected by trypsinisation, and their fluorescence was measured in a flow cytometer (CytoFLEX, Beckman, CA, USA).

In parallel, collected media were centrifugated to remove particulates, and the VEGF level in the supernatant samples was quantified by EliKine™ Mouse VEGF ELISA Kit (Abbkine, Shanghai, China) based on the supplier’s protocol. 

### 2.8. Wound Healing and Transwell Invasion Assays

To assess cell migration, B16F0 cells were cultured in a 12-well plate for 24 h to grow to 70–80% confluency. Then, a 100 µL pipette tip was used to make a scratch, and the scratched cells were washed away with PBS three times. The plated cells were then incubated in media with ABZ-LCP, OTS-LCP or OTS-ABZ-LCP for 4 h, followed by imaging in an Olympus IX81 microscope for cells to migrate to the scratched area. The relative percentage (%) of migrating cells was determined by counting the number of cells migrating to the scratched area compared to that of cells normally cultured in media without any nanoparticles. 

To examine cell invasiveness, B16F0 cells suspended in DMEM without serum were seeded in the upper chamber of Transwell^®^ inserts (8 µm pore size; Corning, NY, USA), while the lower chamber contained DMEM with 10% FBS as the chemoattractant. After incubation with ABZ-LCP, OTS-LCP or OTS-ABZ-LCP at 37 °C with 5% CO_2_ for 24 h, the cells remaining on the upper surface of the membrane were removed gently. The cells migrating to the bottom surface of the membrane were washed with PBS and stained with trypan blue, and the cell number was counted and compared to that with the blank treatment to determine the relative invasive cell percentage. 

### 2.9. In Vivo Tumour Growth Inhibition

The in vivo experiments were carried out according to the Animal Ethics Committee of The University of Queensland guidelines (AIBN/224/18). Six-week old female mice (C57BL/6) were used to establish the tumour model by subcutaneously inoculating 1 × 10^5^ B16F0 cells in DMEM to the left flank of mice. When the tumours grew to 50–100 mm^3^, PBS, ABZ-LCP, OTS-LCP or OTS-ABZ-LCP (equivalent 5 mg/kg ABZ and/or 2 mg/kg OTS964) formulations were injected intraperitoneally (i.p.) at days 1, 3, and 5 into each mouse, and there were 7 mice in each group. 

To evaluate the enhanced in vivo antitumour efficacy by targeted delivery, another animal study was conducted via intraperitoneal injection of PBS, OTS-ABZ-LCP, OTS-ABZ-LCP-P160 and OTS-ABZ-LCP-F100P160, all with equivalent 2.5 mg/kg ABZ and/or 1 mg/kg OTS964, at days 1, 3, and 5 into each mouse, similarly to the above test. 

The tumour size and body weight were measured daily with a digital calliper and balance, respectively. The tumour volume was estimated using ½ × length × width × width. At day 7, these mice were euthanised due to tumour overgrowth in the control group, which was larger than the permitted size, or there was ulceration in 1–2 cases. 

The harvested tumours were digested by a cocktail of collagenase (1.5 mg/mL)/dispase (2 mg/mL). Then, excessive media were added to cease the digestion. The resultant suspension was passed through 70 µm strainers (Falcon) to achieve single cell suspension. The cell number was then adjusted to roughly 10^6^ cell/mL, and the cells were stained with FITC anti-CD4 (Biolegend), anti-CD8/Cy3-goat antirat IgG (Biolegend) or PE anti-PD-L1 (BioLegend), followed by flow cytometry analysis (CytoFLEX, Beckman, CA, USA). 

### 2.10. Statistical Analysis

All relevant data, expressed as mean ± SD, were analysed by *t*-test using GraphPad 7.03 software. The *p* value < 0.05 was considered statistically significant. *p* < 0.05: *; *p* < 0.01: **; *p* < 0.001: ***: *p* < 0.0001: ****.

## 3. Results and Discussion

### 3.1. Physicochemical Properties of LCP-Based Nanoparticles

As shown in Figure 1A, the synthesised ABZ-LCP NPs were spherical in shape. Their average size was 50.7 ± 5.2 nm (Figure 1D), and incorporation of OTS964 (OTS-ABZ-LCP) just slightly increased the average size to 58.8 ± 4.2 nm (Figure 1B,D). Conjugation of two targeting ligands (50–200 folates and 40–160 PD-L1 antibodies) further increased the nanoparticle size by 5–10 nm (Appendix A, Figure 1C,D). However, slight aggregates were observed, as evidenced by a minor peak at 295 nm (Figure 1C,D), probably due to the interactions between surface ligands. The detailed size of OTS-ABZ-LCP NPs conjugated with various concentrations of folic acid and bi-anti-PD-L1 is summarised in Appendix A. Thus, fabricated dual-targeting OTS-ABZ-LCP NPs seem ideal for in vivo applications to minimise the immune clearance.

The zeta potential of ABZ-LCP, OTS-ABZ-LCP, OTS-ABZ-LCP-F50 and OTS-ABZ-LCP-P160 NPs was −19.1 ± 6.4, −17.3 ± 3.2, −14.1 ± 4.5 and −21.6 ± 6.1 mV, respectively. These zeta potential alterations were similar to those reported for FA-conjugated LCP NPs [25] and bi-anti-PD-L1 conjugated LCP NPs [26]. The negative surface charge on the NP surface enhances the blood biocompatibility, decreases the chance of clearance by the reticuloendothelial system (RES), and thus improves NP accumulation at the tumour site [27]. 

The encapsulation efficiency of ABZ in calcium phosphate (CaP) cores was nearly constant for all ABZ-LCP NPs. The composition analysis of ABZ-LCP, OTS-LCP and OTS-ABZ-LCP NPs revealed that the mass percentage of ABZ and OTS in relation to the CaP mass was 25% and 1.5%, corresponding to the encapsulation efficiency of around 60% and 40%, respectively. The loaded amounts of ABZ and OTS964 in LCP NPs could be varied between 0–50% and 0–5.0% in terms of mass, respectively, which were high enough for all subsequent in vitro and in vivo tests.

### 3.2. Cytotoxicity of LCP-Based Nanoparticles against Skin Cancer Cells

Figure 2A,B show the cytotoxic effect of ABZ and OTS964 in the free form (commercial) and the LCP form on B16F0 cells. Firstly, LCP NPs as biodegradable nanocarriers did not cause any cell death in the test concentration range (0–5 µg/mL of ABZ, Figure 2A and 0–130 ng/mL OTS, Figure 2B, which corresponded to 0–20 µg/mL of CaP). Secondly, both free ABZ and OTS964 suppressed cancer growth in a dose-dependent manner. Based on these data, the estimated IC_50_ values of free ABZ and OTS964 were >10 µg/mL and 150 ng/mL, respectively, similar to the values reported for ABZ on MCF 7 cells (IC_50_ = 10–50 µM, i.e., 2.65–13.25 µg/mL) [28] and OTS964 (used to halve the colony size) for U87 or U251 glioma cells (300 nM, i.e., ~130 ng/mL) [29].

ABZ and OTS964 in the LCP form showed higher toxicity against B16F0 cells at the equivalent concentration (Figure 2A,B). Their IC_50_ values were all significantly reduced, being 3.0 µg/mL and 70 ng/mL, respectively. The enhanced cytotoxicity was very similar to our previous report for ABZ-LCP [10], which is attributed to the increased solubility of ABZ in the LCP form [10] and facilitated cellular uptake [22]. Previously, we explained the enhanced solubility and pH sensitivity of LCP NPs, releasing 60–80% of ABZ after 4–48 h at pH 5 compared to 29–52% at pH 6 and 17–29% at physiological pH 7.4 [10].

Based on the IC50 of individual therapeutics in the LCP form, OTS-ABZ-LCP NPs were prepared with the ABZ/OTS mass of 40:1, and more efficiently induced apoptosis of B16F0 cells (Figure 2C). At the drug dose (2.5 µg/mL of ABZ and/or 64 ng/mL of OTS), ABZ-LCP, OTS-LCP and OTS-ABZ-LCP nanoformulations reduced the cell viability to 62.8% ± 3.3%, 65.1% ± 2.3%, and 29.7% ± 3.3%, respectively, revealing that OTS964 and ABZ in the LCP form generated a moderately synergistic cytotoxic effect (62.8% × 65.1%/29.7% = 1.37) on melanoma cells [24]. This observation was similar to that reported by Sugimori et al., where OTS964 and temozolomide generated a synergistic effect on heterogeneous glioma stem cells [29]. As reported, OTS964 is a TOPK inhibitor, and TOPK is overexpressed in many types of human cancers, such as B16 [30], while its inhibition suppresses the stemness of cancer cells. On the other hand, ABZ acts as an antiparasitic and anticancer agent through binding to the β site of tubulin and inhibiting tubulin polymerisation [31]. Their combination thus synergistically enhances cytotoxicity through complementary apoptotic pathways [32]. 

Interestingly, none of the drugs and their LCP formulations showed any significant toxicity against HUVEC cell line as a representative of healthy cells (Figure 2D) at the tested doses, which is consistent with our previous study that there was no significant toxicity against healthy cell line HEK293T from free ABZ or ABZ-LCP [10]. ABZ is widely used as a safe antiparasitic/antiworm drug at the daily oral dose of 10–15 mg/kg [33], and OTS is reported to be safe in the liposomal form [30]. Being safe to healthy cells but toxic to cancer cells is an invaluable advantage over many other chemical anticancer drugs that are toxic to both cell types [34]. 

### 3.3. ROS-Induced Apoptosis, Inhibition of Tumour Progression and Dissemination 

As illustrated in Figure 3A and Appendix A, ABZ-LCP, OTS-LCP and OTS-ABZ-LCP NPs mainly induced late apoptosis in B16F0 cells. At these doses (2.5 µg/mL of ABZ and/or 64 ng/mL of OTS964), ABZ-LCP and OTS-LCP treatment enhanced late apoptosis to 31.6% ± 5.2% and 13.3% ± 3.4%, respectively, significantly higher than 1.8% ± 1.8% for control cells. Even more significantly, OTS-ABZ-LCP treatment resulted in 66.6% ± 4.9% late apoptosis, further showing the synergistic induction of apoptosis via complementary mechanisms of combined ABZ and OTS964 in the LCP form. OTS-ABZ-LCP treatment induced 9.5% ± 1.7% early apoptosis as well. 

The synergistic induction of apoptosis upon OTS-ABZ-LCP treatment may be largely attributed to stimulated ROS generation in B16F0 cells (Figure 3B). Clearly, both single drug (ABZ-LCP and OTS-LCP) treatments enhanced the ROS level from 1.5% ± 0.5% to 13.4% ± 1.3% and 10.7% ± 0.9%, respectively. The combination of ABZ and OTS in the LCP form led to significantly higher ROS production (20.2% ± 1.5%). It is reported that ABZ treated helminthiasis through ROS generation [35], and TOPK inhibition provoked H_2_O_2_-induced oxidative stress [36]. Thus, ROS stimulation plays the major role in synergistic apoptosis induction by the ABZ-OTS-LCP formulation.

Furthermore, LCP-based formulations significantly suppressed tumour proliferation through inducing morphological changes and inhibiting VEGF secretion of cancer cells (Figure 4). As shown in Figure 4A, the B16F0 cell morphology was clearly changed upon treatment with ABZ-LCP, OTS-LCP and OTS-ABZ-LCP NPs. In the ordinary cancer cells, microtubules (arrowed in Figure 4A) extended the cells along the major axis during interphase [37]. The disappearance of spindle elongation in treated cells probably resulted from inhibition of tubulin polymerisation. In addition to morphological changes, a significant reduction of VEGF secretion (by 35–42%) by cells treated with ABZ-LCP and OTS-LCP (Figure 4B) was observed, similar to the VEGF reduction by SKOV-3 cells upon treatment with ABZ [31]. More pronounced inhibition of VEGF secretion (by ~57%) with OTS-ABZ-LCP treatment was observed (Figure 4B), demonstrating again the synergistic actions of ABZ and OTS964 in the LCP form.

Similarly, invasion and migration of B16F0 cells, the key features distinguishing malignant from benign ones [38], were much more significantly inhibited by the LCP-based formulations. As indicated in Figure 4C,D, both ABZ-LCP and OTS-LCP NPs considerably reduced the invasive and migratory activity of skin cancer cells. Note again that OTS-ABZ-LCP NP treatment resulted in a further significant reduction compared with both single drug treatments. As reported previously, the migration of HNSCC [39], Cal27 and SCC154 [39] cell lines was suppressed upon ABZ treatment. Since TOPK overexpression promotes cancer cell clonogenic activity [40] and induces cancer cell migration [40], employing TOPK inhibitor OTS964 along with ABZ thus inhibited tumour progression and metastasis, considering that metastasis starts from invasion of tumour cells into the stroma, followed by migration toward the blood stream [41].

### 3.4. Efficient In Vivo Combination Chemotherapy

Antitumour efficacy of ABZ-LCP, OTS-LCP and OTS-ABZ-LCP NPs was further evaluated in the B16F0-bearing mouse model. After the tumours grew to 50–100 mm^3^, these mice were randomly classified into four groups. Each group was intraperitoneally (ip) injected with PBS, ABZ-LCP, OTS-LCP or OTS-ABZ-LCP nanoformulations containing the equivalent amount of 5 mg/kg ABZ and/or 2 mg/kg OTS 3 times every 2 days (Figure 5A). As displayed in Figure 5A, B16F0 tumours progressed very aggressively in 7 days, and ABZ-LCP treatment reduced the tumour size by almost 50% at day 7, comparable to the previous report for B16F0 [10] and for A549/T tumours in similar conditions [42]. OTS-LCP treatment also reduced tumour size to 50%. This dose is much lower, in sharp contrast with the recent report that six injections of 40 mg/kg OTS (overall dosage of 240 mg/kg) resulted in complete regression of LU-99 lung tumours in five of six mice [34]. 

More interestingly, the combination of ABZ and OTS in the LCP form significantly inhibited tumour growth (by ~85%) compared to the single-drug therapy (~50%) at day 7. As demonstrated in Figure 5A, the treatment with OTS-ABZ-LCP NPs inhibited B16F0 tumour growth. However, the tumour size (152 ± 40 mm^3^) at day 7 was much larger than the initial one (78 ± 30 mm^3^, at day 1), which means that the tumour tissue still grew in some way. Further enhancing the in vivo treatment efficacy as well as minimising chemotherapy side effects may require new approaches, where dual targeting would be a feasible option, as confirmed in the subsequent research presented below. 

### 3.5. Dual Targeting Further Enhances Cytotoxicity against Skin Cancer Cells 

The selected dual targeting ligands were bispecific PD-L1 antibody (bi-anti-PD-L1) and folic acid (FA), which were readily conjugated to the LCP surface [43], which fantastically enhanced the cytotoxicity against B16F0 cells and inhibited the B16F0 tumour growth. As depicted in Figure 6A, targeted delivery by bi-anti-PD-L1-conjugated LCP NPs significantly improved the anticancer effect of OTS-ABZ-LCP NPs via enhanced cellular uptake (Appendix A). B16F0 cell viability was reduced as the number of PD-L1 antibodies on the LCP surface was increased (Figure 6A). The selective targeting capability of bi-anti-PD-L1 of LCP NPs for PD-L1 receptors on the cancer cell surface was confirmed by blocking the PD-L1 receptors with excess bi-anti-PD-L1 before OTS-ABZ-LCP-P160 treatment, where the cell viability (71.0% ± 2.9%) was back to that upon treatment with non-targeting OTS-ABZ-LCP NPs (77.4% ± 4.5%, Figure 6A). This delivery targeting specificity is similar to the report for PD-L1-targeted polyethylene glycol-poly(ε-caprolactone) (PEG-PCL) NPs [26]. 

Similarly to bi-anti-PD-L1 [25], enhanced cellular uptake was confirmed by cytotoxicity study of FA-conjugated LCP NPs (Figure 6B). Conjugating 50 and 100 FA (per LCP) reduced cell viability to 65.6% ± 3.2% and 51.7% ± 5.4%, respectively, compared to 75.1% ± 2.9% upon treatment with non-targeting OTS-ABZ-LCP. The selective targeting for folate receptors by OTS-ABZ-LCP-FA NPs was also confirmed by blocking folate receptors with excess folic acid (Figure 6B).

To develop more efficient dual targeting LCP NPs, several combinations of bi-anti-PD-L1 (40, 80 or 160) and folic acid (50 or 100) were dually conjugated to OTS-ABZ-LCP NPs. As demonstrated in Figure 6C, OTS-ABZ-LCP-F50P160 and OTS-ABZ-LCP-F100P160 NPs significantly reduced cell viability (Appendix A). Cell viability of 29.8% ± 3.2% with OTS-ABZ-LCP-F100P160 at such low drug doses (ABZ: 100 ng/mL and OTS964: 50 ng/mL) is considered a great achievement, as this efficacy can only be achieved at much higher drug doses (ABZ: 2.5 µg/mL and OTS: 64 ng/mL, Figure 2C) using non-targeting OTS-ABZ-LCP NPs. Such significantly reduced drug doses can considerably reduce systematic side effects, a determinant challenge of chemotherapy [25].

### 3.6. Dual Targeting Completely Inhibits Skin Tumour Growth

The in vivo antitumour efficacy of dual targeting two-therapeutic-loaded LCP NPs using half of the previous dosage (2.5 mg/kg of ABZ and 1.0 mg/kg OTS administered 3 times, Figure 7A) was finally confirmed in the B16F0-bearing mouse model. As represented in Figure 7A, the B16F0 tumour size was increased slowly in the OTS-ABZ-LCP-treated group, while 75% tumour size reduction was slightly less than that in Figure 5A (85%) due to the half doses of the two therapeutics used. Interestingly, OTS-ABZ-LCP-P160 NP treatment reduced the tumour size by 85%, similar to that using double therapeutic doses of non-targeting OTS-ABZ-LCP NPs and demonstrating the PD-L1-targeted efficacy. Very powerfully, dual ligand-conjugated OTS-ABZ-LCP-F100P160 NPs completely suppressed the tumour growth and reduced the tumour size by 93%, which was even smaller than the initial size (65 ± 21 mm^3^ at day 7 compared to 78 ± 30 mm^3^ at day 1). The high in vivo efficacy of dual targeting OTS-ABZ-LCP NPs is largely attributed to the over-expression of PD-L1 and FA receptor by B16F0 cells. As displayed in Figure 7B and Appendix A, nearly 50% of B16F0 cells over-expressed PD-L1, which iss expected to enhance the targeting delivery in vivo together with folate receptors. This is similar to the report by Tang et al. that a significant enhancement of payload delivery to human breast tumour overexpressing folate and epidermal growth factor (EGFR) receptors is achieved by dual-targeting LCPs [25]. 

More interestingly, all OTS-ABZ-LCP treatments improved the tumour immune microenvironment (TIME). As represented in Figure 7C,D, both CD4^+^ and CD8^+^ immune cells in the tumour tissues were increased by all OTS-ABZ-LCP treatments. Such an increase indicated that the antitumour immunity is facilitated, probably attributable to neutralisation of the TIME acidity by LCP NPs. While many studies have confirmed the tumour-derived acidity as a major factor for immune suppression [44], LCP NPs may increase the TIME pH (Appendix A) probably via LCP NP degradation through the following reaction [9]:CaHPO_4_ (s) + H^+^ → Ca^2+^ + H_2_PO_4_^−^(1)
which also triggers the release of ABZ and OTS964 in the endosome/lysosome for inducing cancer cell apoptosis. Therefore, the pH elevation may improve the chance of remodelling antitumour immune responses, which is more efficacious in the case of targeted delivery due to enhanced NP accumulation in the tumour tissue [14].

Note that in all groups, no obvious weight loss of mice was observed (Figure 5B and Appendix A), which is consistent with negligible cytotoxicity against HUVEC (Figure 2D) and HEK293T [10] cells and confirms the biosafety of all LCP-based nanoformulations.

## 4. Conclusions

LCP NPs offer a feasible platform for co-encapsulation of hydrophobic drugs in two different ways, i.e., embedded in the CaP cores and in the lipid bilayer. Employing this platform for co-delivery of hydrophobic drugs such as ABZ and OTS964 not only synergistically inhibited tumour cell proliferation through ROS generation and apoptosis induction, but also potentially overcame multidrug resistance and metastasis. Furthermore, co-delivery of ABZ and OTS964 by LCP NPs reduced VEGF expression to suppress angiogenesis and inhibited the invasive and migratory activity of tumour cells while being safe to healthy cells. Moreover, conjugating bispecific PD-L1 antibody and folic acid to OTS-ABZ-LCP NPs further enhanced therapeutic efficacy and minimised side effects as a novel targeting approach. The dual targeting strategy led to a very efficient inhibition at very low drug dosages both in vitro and in vivo, and induced a certain level of immune response. Overall, this platform provides an effective therapeutic strategy, and further studies on the immunotherapeutic potential of the system may open a new window towards a new chemoimmunotherapeutic approach.

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
