# Peer review of "PD-L1-Targeted Co-Delivery of Two Chemotherapeutics for Efficient Suppression of Skin Cancer Growth"

_pharmaceutics, 2022, doi:10.3390/pharmaceutics14071488_

Round 1

Reviewer 1 Report

Title: PD-L1-targeted co-delivery of two chemotherapeutics for effi- 2 cient suppression of skin cancer growth

Manuscript ID: pharmaceutics-1793164

The authors have developed a nanoparticle system called as- lipid-coated calcium phosphate (LCP) encapsulating albendazole (ABZ) and a TOPK inhibitor, OTS964, for skin cancer treatment. The authors observed that co-delivery system (OTS-ABZ-LCP) showed a synergistic cytotoxicity against skin cancer cells through their specific cancerous pathways, without obvious toxicity to healthy cell line. Moreover, dual-targeting the programmed death ligand-1 (PD-L1) and folate receptor overexpressed on the surface of skin cancer cells completely suppressed the skin tumour growth at low doses of ABZ and OTS. The authors concluded that- ABZ and OTS co-loaded dual-targeting LCP NPs represent a promising platform with high potentials against complicated cancers where PD-L1/FA dual targeting appears as an effective approach for efficient and selective cancer therapy.

The authors have perfectly presented the research with perfect description for the introduction section and the discussion section. The results have also been explained in an excellent manner. The reviewer would like to suggest the following minor changes to further improve the manuscript.

Minor comments:

1. In the abstract, “The co-delivery system (OTS-ABZ-LCP) showed” please define- OTS-ABZ-LCP

2. In figure 2-6 Please include the number of experiments “N” for each of these studies. N should be 3 for most of the studies to show statistical significance.

Author Response

Comment 1. In the abstract, “The co-delivery system (OTS-ABZ-LCP) showed” was changed to “OTS and ABZ-loaded LCP (OTS-ABZ-LCP) showed” (Line 12).

Comment 2. Number of experiments (n=3-5) was added to Figures 2-6 captions (Lines 313, 352, 367, 410, 450).

Reviewer 2 Report

The manuscript describes a nanocarrier for codelivery of repurposed albendazole and a novel TOK inhibitor. Overall, the biological part is thorough, whereas the physicochemical characterization of the system is lagging behind.

Comments:

1. M&M section

Treatments are not specified in the transwell invasion assay, so it is not currently understandable.

2.9. Not understandable if 3 times each day, which makes a total of 9 doses. Moreover, the number of animals per group must be specified in both.

2, Release kinetics of both drugs must be explored to gain some insight into the actual dose that is codelivered.

3. Figure 1: TEM images should have the same scale for comparison purposes. Moreover, given the aggregtion peak, further insight must be gained concerning the volumen and/or intensity distribution.

4. 3.2. The equivalent sentence to "Firstly, LCP NPs as biodegradable nanocarriers did not cause any cell death in the test concentration range (0-5 µg/ml of ABZ, Figure 2A, corresponding to 0-20 µg/ml of CaP)." for OTZ  is not said for OTS, so it is certainly not clear if the same concentrations of LCP NPs were tested in both Fig 2A and 2B.

5. Figure 3: early and late apoptosis should be defined in terms of the markers used to characterize them in each case.

6. The sentence "Increasing the FA per LCP NP to 200 instead elevated the cell viability to 61.3% ± 2.2%." should be removed as it is conflictive. There does not seem any statistically significant difference.

7. The following sentence is controversial "PD-L1 antibody conjugates seemed more efficacious than FA in targeting delivery to B16F0 cell as the cell viability caused by OTS-ABZ-LCP-P160 (41.2% ± 2.3%) is significantly less than that (51.8% ± 5.4%) by OTS-ABZ-LCP-F100", as the number of ligands is not the same.

8. 3.6. Why was the expression of folate receptor not evaluated?

9. The effect on TIME is not a matter of targeting in that case, indeed, so it should be clearly stated, so otherwise seems to be misleading.

Author Response

Comment 1. Treatments are now addressed as “After incubation with ABZ-LCP, OTS-LCP or OTS-ABZ-LCP at 37°C with 5% CO2 for 24 h, the cells remaining on the upper surface of the membrane were removed gently” (Line 229).

Comment 2. The number of animals per group was specified for both experiments and the sentence was changed to “formulations were injected intraperitoneally (i.p.) at day 1, 3, and 5 to each mouse and there were  7 mice in each group.” (Line 240-241 and 245).

Comment 3. Release studies were carried out in our previous papers:

Wu, Y.; Gu, W.; Tang, J.; Xu, Z.P. Devising new lipid-coated calcium phosphate/carbonate hybrid nanoparticles for controlled release in endosomes for efficient gene delivery. J. Mater. Chem. B 2017, 5: 7194–7203.

Movahedi, F.; Wu, Y.; Gu, W.; Xu, Z.P. Nanostructuring a widely used anti-worm drug into the lipid-coated calcium phosphate matrix for enhanced skin tumour treatment. ACS Appl. Bio Mater. 2020, 3:4230–4238.

Comment 4. In Figure 1A, the TEM image was replaced with one having the same scale. Thus, all images have the same scale now.

Comment 5. The sentence was changed to “Firstly, LCP NPs as biodegradable nanocarriers did not cause any cell death in the test concentration range (0-5 µg/ml of ABZ, Figure 2A and 0-130 ng/ml OTS, Figure 2B, which is corresponding to 0-20 µg/ml of CaP)” to make it clearer (Line 296).

Comment 6. Annexin V-FITC/PI double staining analysis of apoptosis was demonstrated in Supporting Information (Figure S1), and live, early apoptosis, late apoptosis and necrosis were distinguished by the marker.

Comment 7. The sentence was deleted.

Comment 8. Since the number of folate and PD-L1 antibody ligands were not similar, the comparison of ligand efficacy was removed from the manuscript (Below Figure 6 caption).

Comment 9. Since folate expression on B16 cells are well examined and reported in a wide range of studies (two ref are listed below), the examination was not repeated in this work.

Qiu Z, Xing L, Zhang X, Qiang X, Xu Y, Zhang M, Zhou Z, Zhang J, Zhang F, Wang M, Wang M, et al: CpG oligodeoxynucleotides augment antitumor efficacy of folate receptor α based DNA vaccine. Oncol Rep. 2017, 37: 3441-3448

Wang L, Li M, Zhang N. Folate-targeted docetaxel-lipid-based-nanosuspensions for active-targeted cancer therapy. Int J Nanomedicine. 2012, 7:3281-94.

Comment 10. The relevant sentences were restated to address the effect of OTS-ABZ-LCP treatment on the TIME: “More interestingly, all OTS-ABZ-LCP treatments improved the tumour immune microenvironment (TIME). As represented in Figure 7C and 7D, both CD4+ and CD8+ immune cells in the tumour tissues were increased in all OTS-ABZ-LCP treatments. Such an increase indicates that the antitumour immunity is facilitated, probably attributed to neutralisation of the TIME acidity by LCP NPs”. (Below Figure 7 caption).

Round 2

Reviewer 2 Report

Comments have been duly addressed.